# Accelerating DSP Applications on a 16-Bit Processor: Block RAM Integration and Distributed Arithmetic Approach

Bharathi M [1,2], Krithikaa Mohanarangam [3], Yasha Jyothi M Shirur [4] and Jun Rim Choi [5,6,*]

[1] Department of ECE, School of Engineering Technology, Mohan Babu University,
    Erstwhile Sree Vidyanikethan Engineering College, Tirupati 517102, India; bharathi.m@vidyanikethan.edu
[2] Research Scholar, BNM Institute of Technology, VTU, Bangalore 560070, India
[3] Symbiosis Institute of Technology, Symbiosis International (Deemed University), Pune Campus,
    Pune 412115, India; krithikaamohan@gmail.com
[4] Department of ECE, BNM Institute of Technology, Banashankari, Bangalore 560070, India;
    yashajyothimshirur@bnmit.in
[5] School of Electronic and Electrical Engineering, Kyungpook National University, Daehak-ro, Buk-gu,
    Daegu 41566, Republic of Korea
[6] School of Electronics Engineering, College of IT Engineering, Kyungpook National University, Daehak-ro,
    Buk-gu, Daegu 41566, Republic of Korea
*   Correspondence: jrchoi@ee.knu.ac.kr

**Abstract:** Modern processors have improved performance but still face challenges such as power consumption, storage limitations, and the need for faster processing. The 16-bit Digital Signal Processors (DSPs) accelerate DSP applications by significantly enhancing speed and performance for tasks including audio processing, telecommunications, image and video processing, wireless communication, and consumer electronics. This paper presents a novel technique for accelerating DSP applications on a 16-bit processor by combining two methods: Block Random Access Memory (BRAM) and Distributed Arithmetic (DA). Integrating BRAM as a replacement for conventional RAM minimizes timing and critical route delays, improving processor efficiency and performance. Furthermore, the Distributed Arithmetic approach enhances performance and efficiency by utilizing precomputed lookup tables to expedite multiplication operations within the Arithmetic and Logic Unit (ALU). We use the Xilinx Vivado tool, a robust development environment for FPGA-based systems, for the design process and execute the hardware implementation using the Genesys2 Kintex board. The proposed work produces improved efficiency with a cycle per instruction of 2, where the delay is 2.009 ns, the critical path delay is 8.182 ns, and the power consumption is 4 mW.

**Keywords:** 16-bit processor; distributed arithmetic (DA); Block RAM (BRAM); Xilinx Vivado; Genesys2 Kintex

## 1. Introduction

The increased demand for specialized real-time digital signal processing applications has recently driven the development of efficient hardware solutions such as Digital Signal Processors (DSPs) [1]. These processors play a crucial role in various applications and industries, including telecommunications [2], music processing [3], and image processing [4], by efficiently processing data and executing complex algorithms for real-time tasks. However, conventional DSPs have their limitations. One notable limitation is their dependence on traditional Random Access Memory (RAM) for data storage and retrieval, which can introduce timing delays and critical path issues, negatively affecting overall performance and efficiency. To address these limitations, we explore the integration of Block RAM (BRAM), an alternative to traditional RAM, with Distributed Arithmetic (DA) to minimize timing and critical route delays. Incorporating the DA approach into the design of DSPs addresses the limitations of conventional processors. It eliminates timing delays, increases computational efficiency, and improves overall performance. DA significantly accelerates

multiplication operations using precomputed lookup tables, enhancing the processor's effectiveness. Thus, DSPs can handle multiplication tasks swiftly and efficiently, improving their real-time processing capabilities [5]. Consequently, DSPs are better equipped to meet the growing demands of real-time digital signal processing applications across diverse industries. This article [6] describes a novel circuit design that applies the concepts of classical conditioning to neural networks built with memristors, and this innovation finds application in DSP applications. DSP systems can perform tasks such as pattern recognition, noise reduction, and adaptive filtering across various domains, including audio, image, and communications processing. This circuit design enhances DSP systems' ability to handle complex interactions, adapt to changing conditions, and incorporate memory and inhibition mechanisms. Considering the advantage of integrating BRAM with DA to accelerate DSPs, this paper proposes a design employing the BRAM-DA integration technique, rendering it well-suited for DSP applications such as audio and image processing, where efficient multiplication and accumulation operations are performed every day. Filters are employed in audio processing to perform tasks such as equalization, noise reduction, and effects processing. In audio processing, the DA is used for filtering and convolution and speeds up convolution procedures by utilizing precomputed coefficients and distributed storage of intermediate results. This acceleration is crucial for real-time audio applications.

## 2. Existing Works

This section explores previous methods and advancements associated with improving performance in DSP applications using a 16-bit DSP. Our investigation encompasses architecture, memory hierarchies, instruction sets, energy-saving techniques, and hardware and software integration. The focus is on how the 16-bit DSP demonstrates its potential to enhance performance across various applications.

DSPs specialize in signal processing applications [7], featuring tailored architectures with dedicated Arithmetic Logic Units (ALUs), parallel processing capabilities, and efficient data pathways. These processors use specific Instruction Set Architectures (ISAs) designed for signal processing, allowing for the streamlined execution of complex algorithms. They also contain optimized memory systems, including on-chip cache and scratchpad memory, to reduce latency and increase data throughput. SIMD (Single Instruction, Multiple Data) operations and parallel processing are crucial in accelerating computations in data-intensive applications. Effective collaboration between hardware and software teams during the design process ensures efficient software–hardware co-design.

Given the growing significance of DSPs in modern technology, researchers have proposed various solutions to address their challenges and demands. These solutions aim to enhance the capabilities and efficiency of DSPs for a broad range of applications.

Recently, in [1], the work was focused on improving DSP techniques and applications and highlighted the implementation of advanced methodologies, signal processing methods, and the potential benefits of DSP. Yang et al. presented an affordable, high-performance embedded platform for real-time image acquisition and processing using FPGA and DSP technology [5]. In [6], a novel circuit design is presented that applies the concepts of classical conditioning to neural networks built with memristors; this innovation may find application in DSP applications. Han et al. [8] explored deep learning and scientific computing with DSP architecture, proposing the FT-Matrix2 architecture to enhance arithmetic performance and computational precision. Kapoor et al. emphasized the wide range of applications for DSP [9], particularly in wireless communications and radar signal processing. They also highlighted the versatility of the suggested DSP design, making it suitable for automotive applications. To improve system performance and data efficiency, C.H. Gebotys' 2002 introduces a network flow-based strategy to optimize memory bandwidth utilization in embedded DSP core processors [10].

Donghoon et al. [11] designed a robust 16-bit DSP with advanced features, including a 40-bit ALU, a six-level pipeline, and a $17 \times 17$ parallel multiplier. The paper [12] focuses

on enhancing the efficiency of paraunitary filter banks, with potential applications in signal processing and communications. Alqasemi et al. [13], and colleagues designed a flexible FPGA-based processor for real-time imaging, excelling in quick reconfiguration and efficient resource management. In addition, 5G technology brings transformative improvements, benefiting areas such as autonomous vehicles and remote surgery. Mobile applications use DSPs for enhanced processing and multimedia functions. DA [14] is a promising method for efficiency and real-time processing across various domains. These studies aim to develop real-time solutions using DSP technology to advance their respective fields. They also underscore the importance of algorithm adjustments and performance factors to achieve high-speed processing.

### 3. Proposed Methods

The CPU sequentially fetches program instructions from program memory using memory addresses. Faster memory improves instruction access, and BRAM speeds up execution through parallel processing. Efficient data processing in DSP relies on data memory. Advancements in memory will enhance DSP performance in response to the increasing memory requirements of applications. Integrating BRAM into DSP design and utilizing DA in ALU design [15] significantly enhances real-time signal processing. For instance, real-time audio signal processing involves converting analog audio to digital and applying effects such as reverb, equalization, and noise reduction using the DSP's ALU. DA-based ALU design [16] optimizes tasks such as convolution by replacing complex multiplications with precomputed partial products, reducing computational complexity and enabling quick processing for functions such as filtering and adaptive filtering [17]. The synergy of BRAM and a DA-driven ALU enhances performance, achieving high-quality audio effects and improvements across applications with demanding latency requirements. Incorporating BRAM into the DSP architecture [18] addresses concerns about storage space, power consumption, and efficient data access. This integration considers the following characteristics and factors: 1. Enhanced Data Storage: BRAM enables the storage of more significant amounts of data than distributed RAM, resulting in faster data access and reduced response times. It also maximizes memory utilization. 2. Read and Write Ports: The DSP [19] architecture typically includes two read ports and two write ports for each BRAM, facilitating concurrent memory access and providing flexibility in configuration. 3. Reading and Writing Functionality: BRAM outputs can read or write data based on the status of the write enable pin (wep), enhancing flexibility for data access and modification. 4. Clock Edge Timing: Synchronized read and write operations within BRAM require a clock edge to ensure precise sequencing and efficient data handling.

The proposed DSP, depicted in Figure 1, enhances performance by integrating BRAM into the architecture. This utilization offers benefits such as the ability to read during write operations, multiple read and write ports, and increased data storage capacity, all contributing to the enhanced performance of the DSPs. This paper employs an instruction set to create program instructions customized for a particular processor. These instructions undergo decoding within an instruction decoder, generating the necessary control signals for executing the operations specified in those instructions. For DSP applications, it is essential to note that the instruction set architecture (ISA) differs remarkably from that of conventional processors. Efficiently transferring information between registers and memory is paramount in DSP, specifically when dealing with streaming data. DSP tasks often involve complex mathematical and logical operations.

To meet these requirements, we design instructions that optimize data transfers, particularly for streaming data, and integrate a distributed arithmetic-based technique into the ALU. This ALU is tailored explicitly for DSP calculations. We have also included DSP-specific instructions in the instruction set. These instructions efficiently perform functions such as complex number arithmetic, vector manipulations, and filtering, thereby simplifying data streaming and parallel processing—vital aspects in DSP applications. Figure 2 illustrates the Instruction Register (IR) within BRAM, where the opcode determines

the nomenclature of the IR, representing the specific operation performed. In Figure 2, "dst" refers to the register that stores the value after completing the operations; the "src1" register acts as the source of the first input, and the "src2" register serves as the source of the second input. In reg mode (0), the immediate select pin imm_sel determines whether the information comes from two registers or directly from values provided by the user. When imm_sel is set to 0, this mode corresponds to using a register, while setting it to 1 denotes the immediate way as imm mode (1).

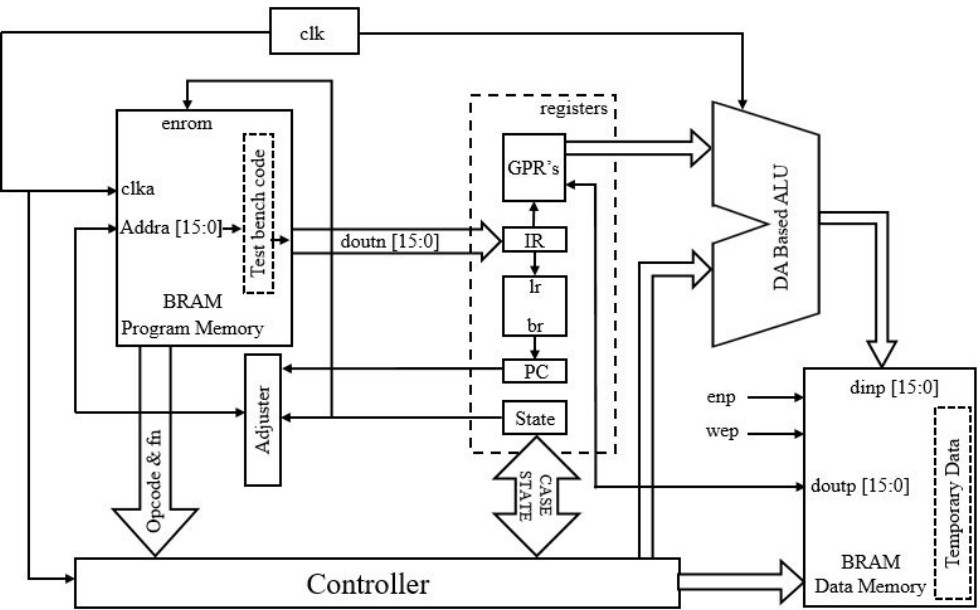

**Figure 1.** Proposed processor's architecture.

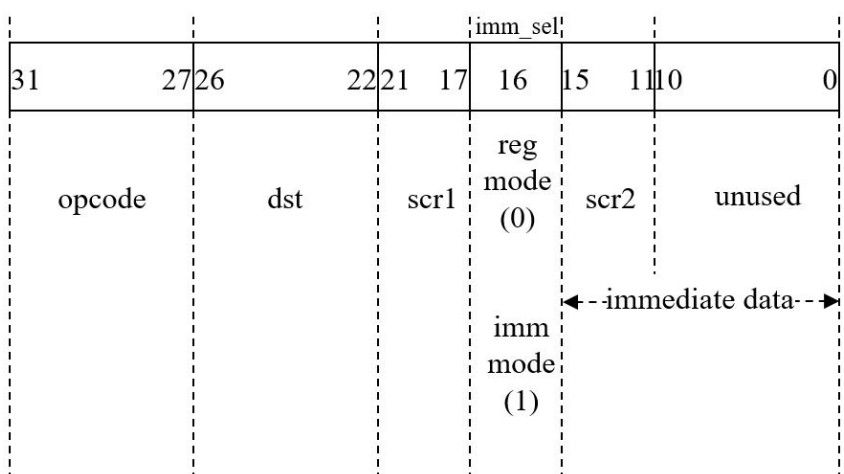

**Figure 2.** A 32-bit Instruction Register.

The IR indicates the opcode using the first five bits, representing the operation performed on the given inputs. This format generates and loads COE files into the program memory. When saving the produced outputs, you need to specify the destination register. Registers store the values to be modified, which you can obtain either from registers or by providing immediate data within the 0–15 bits range. The Instruction Register contains the imm_sel at bit 16, serving this purpose. imm_sel set to 1 uses primary data, while an imm_sel of 0 indicates the usage of values from the source registers. The specified General Purpose Registers (GPRs) store the results of this manipulation. Processors [20] execute software programs by retrieving, decoding, and executing stored instructions from sets. Memory, like BRAM, holds accessible data, and instructions modify and store data with

controlled sequencing. DA Lookup tables assist in specific tasks by boosting arithmetic operations through address-based access. DA processors require fewer gates, employing precomputed tables for efficient computation. Bit-serial DA calculates the inner product in a step, excelling in precise computing. DA eliminates the need for multiplication, focusing on arithmetic. Its efficiency benefits processors by enhancing circuits and ALUs [21], reducing hardware resources for speed and efficiency. In this study, the DA-based approach depicted in Figure 3 increases processor speed by minimizing delays, accelerating DSP performance. Compared to traditional methods, it excels, especially in real-time applications, due to pre-processed data that reduce computational complexity and delays. DA operates bit-serially, reordering multiply and accumulating operations. When used in the ALU of DSP, it simplifies operations and enhances the retrieval of outputs, making the ALU highly effective in signal processing. The DA-based ALU's instruction set comprises 32 instructions: the first 16 are logarithmic, and the latter 16 are arithmetic. It executes operations on filtered bits [22,23] based on user-provided addresses for specific functions.

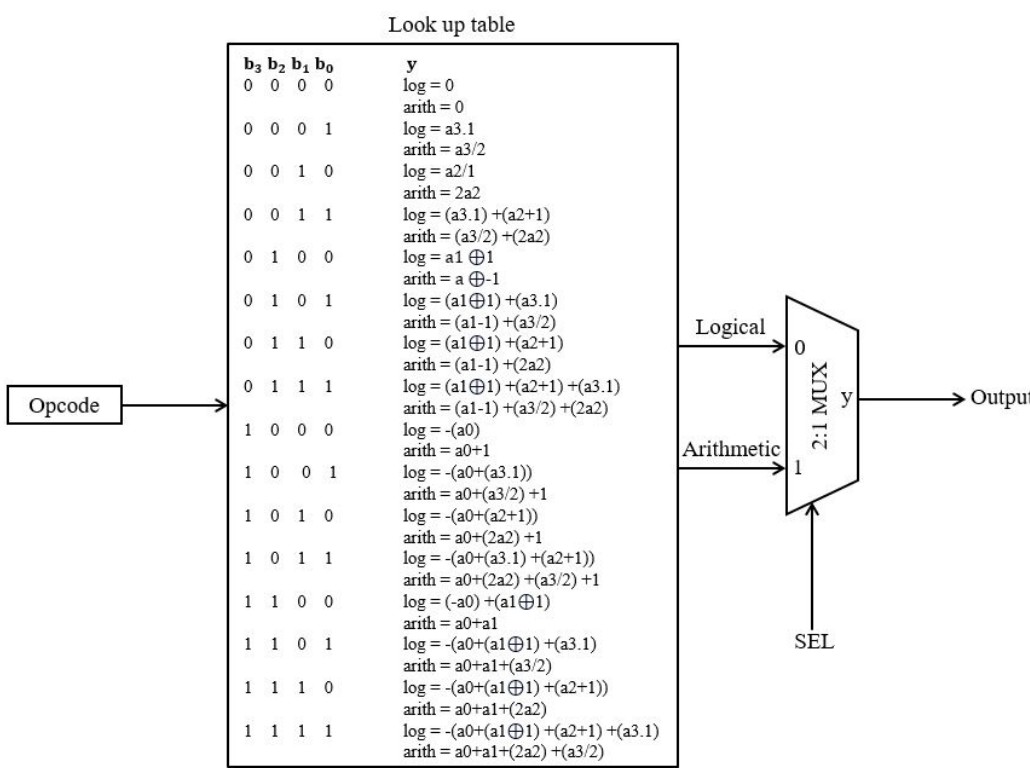

**Figure 3.** DA Look-Up Table.

## 4. Results

The proposed processor's functioning becomes apparent through the following outlined steps. These steps offer insights into how the processor operates and executes tasks based on the provided test bench code. The BRAM program memory has the test bench code, which aligns with the IR structure. Operation initialization: The processor's operation commences when the enable pin of the program memory is 1. Instruction fetching: When enrom = 1, the processor retrieves the next instruction's value from the address [15:0] and loads it into doutn [31:0] within the program memory. Instruction decoding: The test bench code provides instructions and their execution sequence. The IR can access data from general-purpose registers or immediate data based on the instruction's rapid selection. Controller and status register: The controller interprets information from the IR based on the provided opcode. Register data storage: For instructions related to data storage or movement, the controller directly accesses GPRs to carry out the operation.

ALU operations: The controller triggered by instructions requires input procedures to fetch values from GPRs and send them to the ALU. The ALU executes operations on the inputs and temporarily stores results in the data memory, acting as temporary storage. ALU functionality: The ALU accesses data from GPRs, executes instructed operations, and temporarily stores the output in BRAM data memory before permanently updating GPRs. Output observation: You can view the output from the RAM by activating the read enable pin; after using the write, allow the pin to store the work in the RAM. Permanent storage: To permanently store the output in registers, the controller sends data from RAM's doutp to GPRs, storing the result in the specified destination register. Continuous execution: While ALU operations are taking place, the system simultaneously retrieves the following input from the testbench code (COE file), enabling continued instruction execution. Jump and branch handling: When the processor encounters jump or branch instructions, the program counter manages the execution flow, ensuring smooth transitions. Usage of lr and blr registers: The processor employs lr and blr registers for branching and jump operations, allowing it to execute specific instructions. Jump execution: The values in lr and blr registers determine the destination of jumps, temporarily altering the PC to execute instructions accordingly. Controller and status register usage: The controller actively utilizes the status register to determine the operational state, contributing to delay management.

Handling delays: The processor's operation includes a default delay, and to prevent further delay amplification, it introduces a two-cycle delay after each procedure. Synchronizing with the clock: The processor synchronizes its execution using a clock, with operations occurring at the positive edge of each clock cycle. Fetching and executing instructions: After every two-cycle delay, the processor loads the following input into the IR and proceeds with execution based on the opcode, and the program counter increments after each operation. Enhancing speed and reducing delay: The implementation of this processor effectively tackles issues related to speed and delay in execution. The proposed processor efficiently executes tasks according to the test bench code, managing delays, branching, and control flow while utilizing various registers and memory components. This implementation results in improved speed and reduced delay, enhancing overall performance.

To illustrate, we achieved the results using the Xilinx Vivado tool, following the instructions above, demonstrating a simple process of repeatedly subtracting a number using a loop. This process concludes when the value reaches zero. We presented the test bench and algorithmic program code related to this in Figure 4 (top) for reference. In Figure 4 (bottom), we depict the operation where we copy the instruction in doutn [31:0] into IR during the enabling of the program memory (enrom = 1). For the zeroth instruction, each time we update the Instruction Register, the ALU and GPRs execute the instruction and store the resulting data in destination registers through data memory, requiring three delay cycles to activate the subsequent instruction, as illustrated in Figure 5 (top). Lastly, Figure 5 (bottom) shows the sixth instruction—a new instruction process begins each time enrom = 1. Figure 6 illustrates logarithmic and arithmetic values within a DSP based on DA.

Each instruction takes two clock cycles to complete in a fundamental process during simulation operation. The next instruction is fetched and loaded into the instruction register during this cycle. The CPU then decodes the opcode, determining what specific action needs to be carried out. Depending on this opcode, the processor performs various activities, including arithmetic and logical computations, data manipulation, and orchestrating control flow actions. The basis of a processor's functionality is the integration of sequential tasks, which guarantees the timely execution of commands. In this work, we performed simulations for several operations, focusing on the zeroth, first, fourth, and sixth instructions and a halt operation. We consider the total number of operations carried out across all instructions as part of calculating total execution time, even though the article only shows findings for the zeroth and sixth instructions. Our proposed architecture out-

performs conventional DSP by significantly reducing execution time and demonstrating its suitability for accelerated DSP applications.

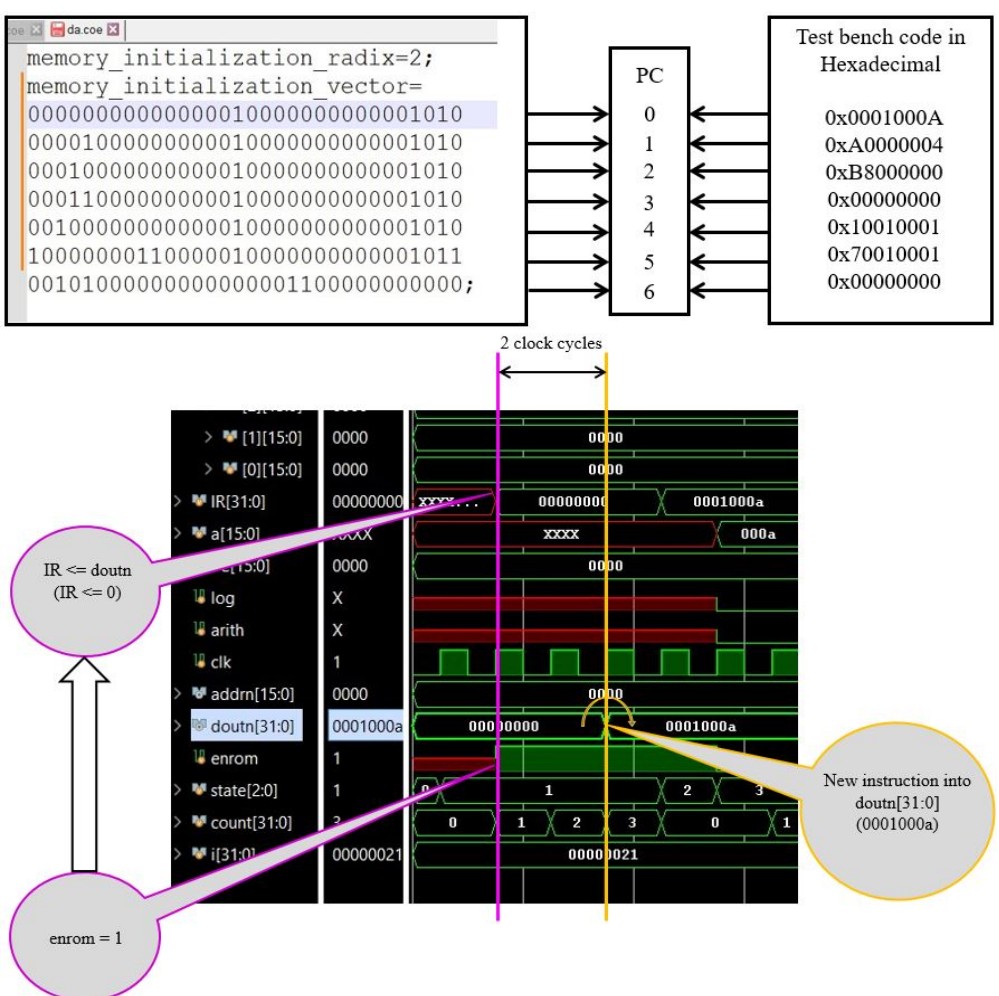

**Figure 4.** Test bench code with DA (**top**); Enabling ROM (**bottom**).

$T_{total}$ for the conventional processor's overall execution and DA-Based DSP Processor are calculated as follows: for conventional processor's overall execution time, $T_{total} = (OverallCycles \times CycleDuration)$. For the clock frequency of 5 MHz, $T_{total} = T_{exec-0} + T_{exec-1st} + T_{exec-4} + T_{exec-6} + T_{Halt}$. i.e., $T_{total} = (3 + 4 + 1 + 4 + 120)/5,000,000) = 26.4$ microseconds. For DA-Based DSP Processor with a clock frequency of 5 MHz, $T_{total} = T_{exec-0} + T_{exec-1st} + T_{exec-4} + T_{exec-6} + T_{Halt}$. i.e., $T_{total} = (2 + 3 + 1 + 4 + 100)/5,000,000) = 22$ microseconds.

We can comprehend the modifications and improvements achieved by integrating this DA-based approach into the DSP. We measure the timing delay at 2.009 ns, the critical path delay at 8.182 ns, and the power consumption at 4 mW. The utilization of the DA technique has notably enhanced the overall performance in speed and delay, leading to substantial reductions. We provide a comparative table showcasing the disparities between a conventional DSP and a DA-based DSP in Table 1. Our implementation results indicate a significant enhancement in the DA-based DSP compared to the conventional processor. We carried out the simulation results using Xilinx Vivado, and we observed crucial metrics, including delay (logic delay and net delay) and power (static and dynamic), which were also presented in Table 1. The proposed processor results in increased performance and efficiency compared to existing approaches.

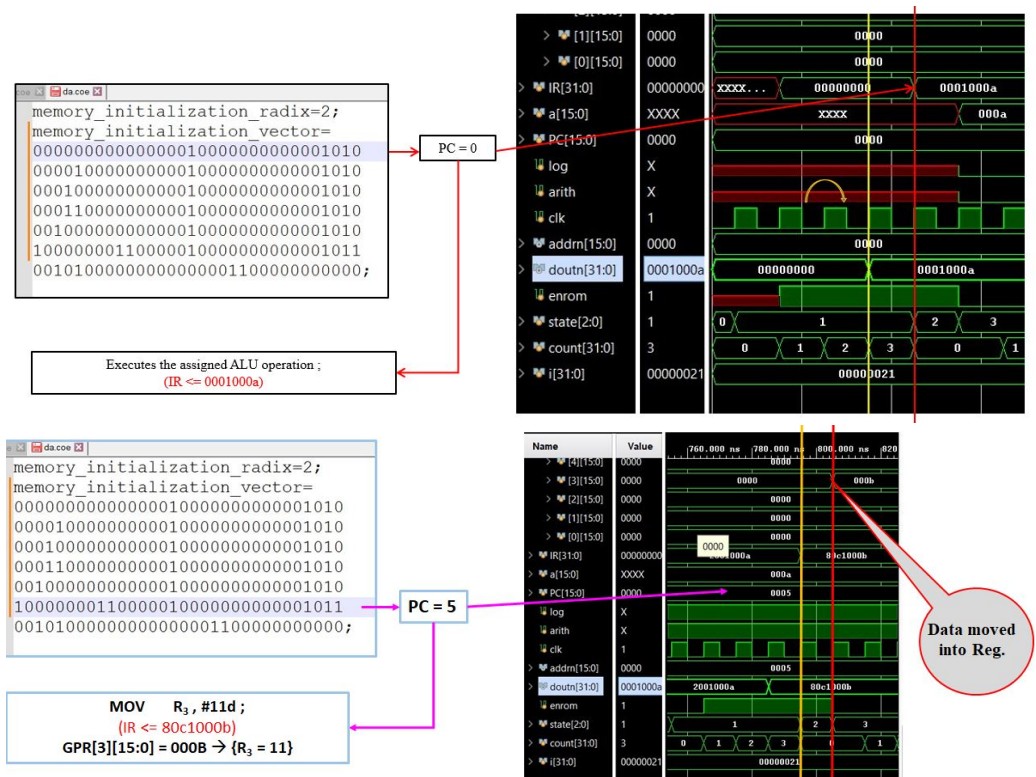

**Figure 5.** Execution of 0th instruction (**top**); Execution of 6th instruction (**bottom**).

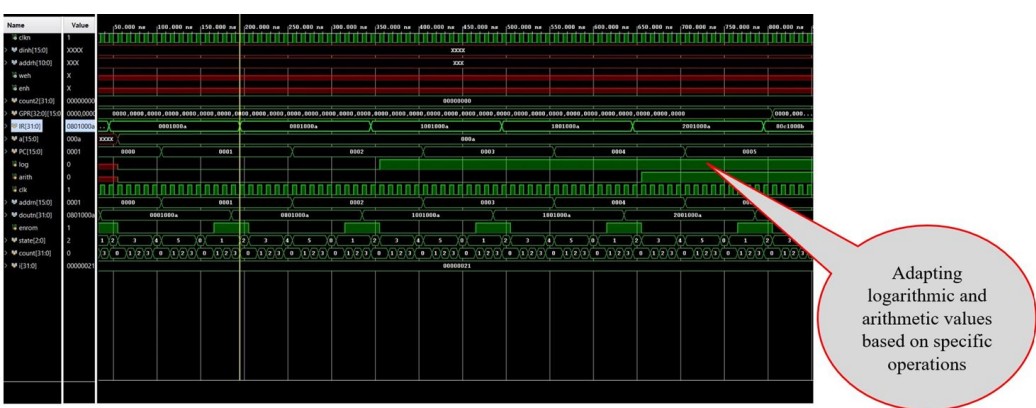

**Figure 6.** Logarithmic and arithmetic values in DA-based DSP.

**Table 1.** Comparison table between 16-bit existing DSP vs. proposed DSP. Power is in W (SP—Static power, DP—Dynamic power, and TP—Total power) and delay is in ns (LD—Logic delay, ND—Net delay, and TD—Total delay).

| Type | On-Chip Power (W) | SP | DP | TP | Critical Path (ns) | LD | ND | TD |
|------|-------------------|------|------|------|--------------------|------|------|------|
| Existing [24] | 0.431 | 0.163 | 0.268 | 0.431 | 8.818 | 2.745 | 1.022 | 3.767 |
| Proposed | 0.301 | 0.162 | 0.139 | 0.301 | 8.182 | 0.423 | 1.586 | 2.009 |

In Table 2, the proposed method is compared to existing 16-bit DSP based on cycles per instruction, instruction word length, vectorization support, and distributed arithmetic. Ref. [25] stands out as a reliable option due to its vectorization capability and 32-bit instruction length, which make it ideal only for Convolutional Neural Networks (CNNs) and real-time processing requirements. In contrast, ref. [26] uses a 32-bit instruction length

but lacks vectorization, focuses on complicated Fast Fourier Transform (FFT) design, and primarily serves high-performance computing applications. With 16-bit instruction length and capabilities for vectorization, ref. [27] aims to be a loop accelerator for real-time applications with high computational time. These three well-known references provide insightful information about processor design, but depending on their intended usage, they also have certain advantages and disadvantages. While focusing on a distributed arithmetic-based DSP that meets both real-time and high-performance computing needs, the suggested method takes a novel approach with a 32-bit instruction length and vectorization support. Our proposed method also offers a flexible processor architecture that has the ability to unite real-time and high-performance computing tasks.

**Table 2.** The comparison of various implementations for the 16-bit Processor (CPI—Cycles per Instruction; IWL—Instruction Word Length; VS—Vectorization Support; DA—Distributed Arithmetic).

| Design | IWL | VS | Suitability of a Processor | Requirements | CPI |
|---|---|---|---|---|---|
| [25] | 32 | yes | Convolutional Neural Networks | Real-time | - |
| [26] | 32 | No | Complex FFT Design | High performance computing | 3 |
| [27] | 16 | yes | Loop Accelerator | Real-time | - |
| Proposed | 32 | yes | DA-based DSP Processor | Real-time + High performance computing | 2 |

## 5. Discussion and Conclusions

The proposed architecture improves the efficiency of DSPs, enabling them to effectively address the demands of contemporary real-time signal processing tasks. When we apply this approach to the Harvard design using BRAM, it notably enhances memory capacity. Using Xilinx Vivado, we demonstrate substantial enhancements in the DSP's implementation, reducing timing delay to 2.009 ns and improving the critical path delay to 8.182 ns. Notably, the primary processor has the drawback of higher on-chip power consumption of 0.431 watts. Nevertheless, this work achieves reduced delay, heightened speed, and enhanced processor performance compared to standard DSPs. The processor's structure suits real-time signal processing, effectively handling single input signals using a DA-based ALU, utilizing precomputed values from lookup tables. Our study focuses on DSPs architecture and highlights its potential for further exploration, which remains significant, particularly for DSPs with larger read-only memory widths that address potential time-consuming operations on 64-bit or 32-bit processors.

To ensure smooth and efficient operation, modern processors employ several complex mechanisms. Among these, status registers are crucial in maintaining control over operations and addressing processing delays. This article delves into the significance of status registers, their impact on performance optimization, and the synchronization mechanism that balances everything. Status registers are integral components of a processor as they reflect the operational state at any given moment. They play a vital role in managing processing delays, employing an efficient method to prevent additional delays from impeding effective execution. This method involves imposing a two-cycle delay and adjusting it after each operation. This approach effectively avoids potential bottlenecks, ensuring that the processor controls its functions. Critical strategies for improving processor performance encompass the utilization of status registers, synchronized clock signals, and the implementation of two-cycle delays. Together, these procedures maintain control, address unforeseen delays, and ensure the execution of instructions in the correct sequence. By comprehending and using these components, processor designers can construct efficient and responsive computing systems that meet the demands of today's technology-driven world. From Table 1, the current processor has a logic delay of 2.745 ns, while the proposed processor achieves a much lower delay of 0.423 ns, indicating a significant speed improvement. The proposed processor has a slightly higher net delay at 1.586 ns than the existing processor's 1.022 ns. However, the proposed processor still has a lower overall delay at 2.009 ns compared to the current processor's 3.767 ns, enhancing the overall processing

time for real-time applications. Regarding power consumption, the static power remains similar between the two processors (0.163 W for the existing and 0.162 W for the proposed). Notably, the proposed processor significantly reduces dynamic power consumption to 0.139 W from the current processor's 0.268 W during active operation, signifying improved power efficiency. The proposed processor also exhibits a lower total on-chip power of 0.301 W compared to the existing processor's higher 0.431 W, highlighting its potential for energy savings and better power efficiency.

Digital signal processing tasks can be efficiently executed using languages and tools such as MATLAB, Python, LabVIEW, C, Verilog, and GNU Octave, each catering to various DSP application domains and hardware requirements. Our proposed DSP architecture is adaptable to these programming languages and tools, allowing seamless integration and optimization for various signal processing applications. Developing software for a completely new DSP architecture is more challenging than modifying an existing one. However, building a DSP architecture from the ground up and customizing it with a technique called Distributed Arithmetic allows developers to tailor it precisely to the specific needs of different applications. Like other works [28–30] involving software working with DSP architecture, our proposed method is also adaptable to various programming languages and platforms. For instance, customizing a DSP architecture with Distributed Arithmetic can significantly improve data rates and efficiency in wireless communication. Similarly, it can enhance the quality of audio processing tasks such as noise reduction and sound improvement. Additionally, custom DSP architectures can accelerate image and video compression methods such as JPEG and H.264, leading to faster encoding and decoding while conserving power.

In conclusion, the proposed 16-bit DSP showcases notable improvements in delay and power parameters compared to the current processor. The reduction in logic delay and total delay in the proposed processor signifies a considerable boost in processing speed and efficiency, making it an excellent fit for real-time applications. Furthermore, the decrease in dynamic power consumption and total on-chip power highlights enhanced power efficiency, potentially leading to energy conservation. These outcomes emphasize the advantages of adopting the proposed processor for real-time signal processing tasks. Future research should focus on extending the evaluation of the proposed DSP architecture in a wider range of real-time signal processing applications, optimizing power efficiency, and exploring scalability to support higher memory widths.

**Author Contributions:** Conceptualization, K.M. and J.R.C.; Validation, B.M.; Formal analysis, B.M.; Investigation, B.M.; Writing—original draft, B.M. and K.M.; Writing—review & editing, K.M., Y.J.M.S. and J.R.C.; Supervision, Y.J.M.S.; Funding acquisition, J.R.C. All authors have read and agreed to the published version of the manuscript.

**Funding:** This work was supported by Samsung Electronics Co., Ltd.

**Data Availability Statement:** The authors declare that the data supporting the findings of this study are available within the paper.

**Acknowledgments:** Samsung Electronics Co., Ltd. supported this work.

**Conflicts of Interest:** The authors have no conflict of interest to declare that are relevant to the content of this article.

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
