# Peer review of "Accelerating DSP Applications on a 16-Bit Processor: Block RAM Integration and Distributed Arithmetic Approach"

_electronics, doi:10.3390/electronics12204236_

Round 1
Reviewer 1 Report (Previous Reviewer 1)
The reviewer had pointed out two serious flaws of the previous version of the manuscript: lack of architectural considerations and incomplete performance evaluation. These were partially improved in the revised version, but the manuscript still needs a major revision in order to be benefitical for potential readers. Thus, the reviewer would like to present three conditions for acceptance as follows.
(1) A reasonable explanation how an application developer writes their own application for the proposed DSP processor should be added. The reviewer now understood the need to design an instruction set from scratch. However, software development for a full-scratch architecture generally becomes more difficult than that for a modified version of existing architecture. A discussion in this viewpoint is missing in the current version of manuscript.
(2) The "conventional" (l. 226), or "existing" (Table 1), DSP has to be clearly defined. Is it a publicly available processor, for example, by being published at GitHub or somewhere? Or, did the authors simply develop the proposed processor without distributed arithmetic? The definition is important to properly understand the evaluation results.
(3) A comparison between the "conventional" DSP and the "DA-based" DSP with the number of cycles taken for the same application is required. As pointed out in the previous comment, the critical path delay is not enough to evaluate the execution time of an application. The execution time is the product of the cycle time and the number of cycles. In order to prove that the proposed architecture is "accelerating DSP applications," as stated in the title, the best way is to compare the execution time of the same application.
Author Response
Response to Reviewer: Thanks for your comments and detailed suggestions. We have revised the paper to address all the issues raised by the reviewer.
Q1. A reasonable explanation of how an application developer writes their own application for the proposed DSP processor should be added. The reviewer now understood the need to design an instruction set from scratch. However, software development for a full-scratch architecture generally becomes more difficult than that for a modified version of the existing architecture. A discussion of this viewpoint is missing in the current version of the manuscript.
A1. Thank you for the comment.
Digital signal processing (DSP) tasks can be efficiently executed using languages and tools like MATLAB, Python, LabVIEW, C, Verilog, and GNU Octave, each catering to various DSP application domains and hardware requirements. Our proposed DSP architecture is adaptable to these programming languages and tools, allowing seamless integration and optimization for various signal processing applications."
Developing software for a full-scratch architecture is typically more challenging than modifying an existing one. However, fully customizing DSP architectures with distributed arithmetic from scratch allows developers to tailor them to unique application needs. In contrast, changing existing architectures can restrict customization. Customizing DSP architectures with distributed arithmetic from scratch is also valuable for specific signal processing tasks like wireless communication and audio processing, improving performance and energy efficiency. Furthermore, tailoring a fully customized distributed arithmetic-based architecture for wireless communication can enhance data rates and efficiency. Custom DSP architectures also benefit high-quality audio processing, such as noise reduction and enhancement. Additionally, DSP architectures can accelerate image and video compression methods like JPEG and H.264, speeding up encoding and decoding while conserving power.
We have also added references to support this discussion.
The discussion suggested by the reviewer is added in the revised manuscript as below.
Digital signal processing (DSP) tasks can be efficiently executed using languages and tools like MATLAB, Python, LabVIEW, C, Verilog, and GNU Octave, each catering to various DSP application domains and hardware requirements. Our proposed DSP architecture is adaptable to these programming languages and tools, allowing seamless integration and optimization for various signal-processing applications.
Developing software for a completely new digital signal processor (DSP) architecture is more challenging than modifying an existing one. However, building a DSP architecture from the ground up and customizing it with a technique called distributed arithmetic allows developers to tailor it precisely to the specific needs of different applications.
Like other works [1] [2] [3] involving software working with DSP architecture, our proposed method is also adaptable to various programming languages and platforms. For instance, customizing a DSP architecture with distributed arithmetic can significantly improve data rates and efficiency in wireless communication. Similarly, it can enhance the quality of audio processing tasks such as noise reduction and sound improvement. Additionally, custom DSP architectures can accelerate image and video compression methods like JPEG and H.264, leading to faster encoding and decoding while conserving power.
[1], R. Gu, J. W. Janneck, S. S. Bhattacharyya, M. Raulet, M. Wipliez and W. Plishker, "Exploring the Concurrency of an MPEG RVC Decoder Based on Dataflow Program Analysis," in IEEE Transactions on Circuits and Systems for Video Technology, vol. 19, no. 11, pp. 1646-1657, Nov. 2009, doi: 10.1109/TCSVT.2009.2031517.
[2] R. W. Robison, "Engineering software-tools for embedding DSP," in IEEE Spectrum, vol. 29, no. 11, pp. 81-82, Nov. 1992, doi: 10.1109/6.166505.
[3]M. Sala, F. Salidu, F. Stefani, C. Kutschenreiter and A. Baschirotto, "Design considerations and implementation of a DSP-based car-radio IF Processor," in IEEE Journal of Solid-State Circuits, vol. 39, no. 7, pp. 1110-1118, July 2004, doi: 10.1109/JSSC.2004.829402.
Q2. The "conventional" (l. 226), or "existing" (Table 1), DSP has to be clearly defined. Is it a publicly available processor, for example, by being published at GitHub or somewhere? Or, did the authors simply develop the proposed processor without distributed arithmetic? The definition is important to properly understand the evaluation results.
A2. Thank you for the comment.
The conventional article [4] presents the MDSP-II, a specialized 16-bit DSP created for mobile communication. To evaluate its performance, we investigate its architecture and do simulations using the Xilinx Vivado tool. To address the reviewer's comment, we have also added the reference to the paper.
[4] Byoung-Woon Kim et al., "MDSP-II: a 16-bit DSP with mobile communication accelerator," in IEEE Journal of Solid-State Circuits, vol. 34, no. 3, pp. 397-404, March 1999, doi: 10.1109/4.748192.
Q3. A comparison between the "conventional" DSP and the "DA-based" DSP with the number of cycles taken for the same application is required. As pointed out in the previous comment, the critical path delay is not enough to evaluate the execution time of an application. The execution time is the product of the cycle time and the number of cycles. In order to prove that the proposed architecture is "accelerating DSP applications," as stated in the title, the best way is to compare the execution time of the same application.
A3. Thank you for the valuable suggestion. The following details are added in the revised manuscript:
Each instruction takes two clock cycles to complete in a fundamental process during simulation operation. The next instruction is fetched and loaded into the instruction register during this cycle. The CPU then decodes the opcode, determining what specific action needs to be carried out. Depending on this opcode, the processor performs various activities, including arithmetic and logical computations, data manipulation, and orchestrating control flow actions. The basis of a processor's functionality is the integration of sequential tasks, which guarantees the timely execution of commands. In my research, I performed simulations for several operations, focusing on the 0th, 1st, 4th, and 6th instructions and a halt operation. I'm considering the total number of operations carried out across all instructions as part of calculating total execution time, even though the article only shows findings for the 0th and 6th instructions. Our proposed architecture outperforms conventional DSP by significantly reducing execution time and demonstrating its suitability for accelerated DSP applications.
The conventional processor's overall execution time,
T_total is calculated as follows:
T_total = (overall Cycles * Cycle Duration).
Clock frequency (f) = 5 MHz (5,000,000 Hz)
T_total_conventional = T_exec_0 + T_exec_1st + T_exec_4 + T_exec_6 + T_Halt.
T_total_conventional = (3/5,000,000) + (4/5,000,000) + (1/5,000,000) + (4/5,000,000) + (120/5,000,000) seconds
T_total = (132/5,000,000) = 0.0000264 = 26.4 microseconds
DA Based DSP Processor:
Clock frequency (f) = 5 MHz (5,000,000 Hz);
T_total_proposed = T_exec_0 + T_exec_1st + T_exec_4 + T_exec_6 + T_Halt
T_total_proposed= (2/5,000,000) seconds + (3/5,000,000) seconds + (1/5,000,000) seconds+(4/5,000,000) +(100/5,000,000) seconds
T_total = (110 / 500,000) = 0.000022= 22 microseconds
Reviewer 2 Report (Previous Reviewer 2)
It is OK now after the last response! It can be accepted in this journal!
ok
Author Response
Thank you for the positive feedback. If you have any additional instructions or requirements, please feel free to inform us.
Round 2
Reviewer 1 Report (Previous Reviewer 1)
The reviewer confirmed that the authors sincerely answered the comments to the previous version and the manuscript becomes improved well. Nice analysis and discussion.
This manuscript is a resubmission of an earlier submission. The following is a list of the peer review reports and author responses from that submission.
Round 1
Reviewer 1 Report
This manuscript presents an architecture of DSP processor that utilizes distributed arithmetic (DA). The authors provided power and delay estimations from the Vivado tool as the evaluation results of their processor. However, architectural considerations for custom processing systems and performance evaluation using target applications are missing in the current manuscript.
Regarding the architectural considerations, reasons for designing an instruction set (ISA) from scratch should be explained at first. The instruction formats presented in Fig. 2 is similar to ones in existing RISC ISA, such as MIPS and RISC-V. The authors could have made DA operations be custom instructions of an existing architecture, which would have been beneficial in the sense of utilizing existing processors and development environments.
Regarding the performance evaluation, not only the critical path delay (or operating frequency), but also the number of instructions and the CPI (cycles per instruction) are required to evaluate the execution time of target application. This means that the instruction set and the microarchitecture have to be properly designed, in addition to the circuit itself. Merits of adopting the proposed architecture in these viewpoints were neither described nor evaluated in the manuscript.
In addition, the "basic" DSP and the "existing" processor, presented in Tables 1 and 2, are not clear. Instead of naive implementations, comparison with well-considered implementations appeared in other studies is highly recommended.
The manuscript is basically comprehensible but the reviewer found some minor errors.
- l. 225: power( static and dynamic) -> power (static and dynamic)
- l. 235: 53 watts -> 53 milliwatts (or 53 mW)
Reviewer 2 Report
In this paper, Block RAM integration techniques and Distributed Arithmetic are combined. Block RAM reduces timing delays and critical route delays. The Distributed Arithmetic speeds up the ALU multiplication operation. A simple computational procedure is shown by Xilinx vivado tool. The better performance of the proposed data-based DSP processor is demonstrated. This work provides some references for improving the efficiency of Digital Signal Processing. Here are some comments.
In the third paragraph of Chapter 2, some recent related research advances are described. In addition, authors are advised to refer to some articles that utilize novel components to improve the efficiency of data processing. For example, Memristor-based neural network circuit with multimode generalization and differentiation on pavlov associative memory. IEEE Transactions on Cybernetics.
In Chapter 4, a comparative table showcasing the disparities between a conventional DSP Processor and the DA-based DSP Processor is provided in Table 1. However, the calculation process of data processing power consumption is not detailed enough. The authors are advised to elaborate further in the text.
In this paper, the working process of the proposed method with the simulation diagram is described. At the same time, the necessary instructions are added. However, some figures have poor clarity, such as Figure 3 and Figure 5 (b). The author try to adjust, please.
Moderate editing of English language required